# Modeling Stratum Corneum Swelling for the Optimization of Electrode-Based Skin Hydration Sensors

**DOI:** 10.3390/s21123986

**Published:** 2021-06-09

**Authors:** Claudio Malnati, Daniel Fehr, Fabrizio Spano, Mathias Bonmarin

**Affiliations:** Institute of Computational Physics, School of Engineering, Zurich University of Applied Sciences, ZHAW, 8400 Winterthur, Switzerland; claudio.mi.malnati@gmail.com (C.M.); daniel.fehr@zhaw.ch (D.F.); fabrizio.spano@zhaw.ch (F.S.)

**Keywords:** skin model, skin hydration, dielectric spectroscopy, stratum corneum

## Abstract

We present a novel computational model of the human skin designed to investigate dielectric spectroscopy electrodes for stratum corneum hydration monitoring. The multilayer skin model allows for the swelling of the stratum corneum, as well as the variations of the dielectric properties under several hydration levels. According to the results, the stratum corneum thickness variations should not be neglected. For high hydration levels, swelling reduces the skin capacitance in comparison to a fixed stratum corneum thickness model. In addition, different fringing-field electrodes are evaluated in terms of sensitivity to the stratum corneum hydration level. As expected, both conductance and capacitance types of electrodes are influenced by the electrode geometry and dimension. However, the sensitivity of the conductance electrodes is more affected by dimension changes than the capacitance electrode leading to potential design optimization.

## 1. Introduction

Dielectric spectroscopy offers the potential of non-invasive monitoring of skin hydration. Commercially available and well-established instruments, such as the MoistureMeterD (Delphin Technology AG), the Corneometer^®^ CM 825 (Courage + Khazaka electronic GmbH), or the Nevisense 3.0 (SciBase AB), are used in dermatology, for analyzing various diseases, and for determining the effectiveness of medical therapies [1,2,3]. Much work has been invested in optimizing electrodes and developing new technologies. Skin-like or drawn-on electrodes have been presented for real-time monitoring and wireless measurements [4,5,6].

A numerical model simulating the dielectric properties of human skin can help to understand the effects of dielectric spectroscopy within the skin and to study the influence of skin hydration. Various publications characterize biological tissues [7,8,9] and report the measurement of dielectric properties of the human skin [10,11,12,13]. A semi-analytic solution for a three-layer model of stratum corneum, viable skin, and adipose tissue is proposed [14]. In [15], a model is presented to calculate the effective dielectric properties for each skin layer from numerical cellular structures parametrized by sub-cellular and cellular compounds.

The outermost layer of skin, the stratum corneum (SC), functions as a barrier to water loss and as a barrier to exogenous chemical penetration. SC water content is known to reflect skin health, and is related to changes in elasticity, flexibility, and surface morphology [16,17,18]. Li et al. presented a mass transport model to predict water transport through the SC with a moving boundary for the swelling of the SC. The model of water transport through the SC involves both diffusive and convective transport and can represent both transient hydration and dehydration of the tissue [19,20].

The objective of this work consists of the development of a reliable, computationally efficient numerical model for the hydration-dependent dielectric behavior of the human skin. A multilayer skin model was set up, each layer characterized with its dispersive dielectric properties, in combination with a SC hydration model. Focusing on the SC layer, the dispersive dielectric parameters were fitted to estimate the dielectric properties under the influence of hydration. To further investigate hydration influences on dielectric spectroscopy, the SC thickness was set up to be hydration-dependent as well.

With the SC hydration model, the range of water concentration was investigated by exposing the skin surface to air at various relative humidities. Three different fringing-field electrodes were employed on the skin model in order to investigate the influence of the hydration-dependent SC thickness on the dielectric spectroscopy. Finally, a sensitivity analysis was performed with the three different fringing-field electrode types with various geometries.

## 2. Materials and Methods

Different physical models are combined to simulate the behavior of the human skin under dielectric spectroscopy measurements. Various work investigated the SC’s hydration influences. The hydration dependency of the dielectric properties is reported in [21], and the SC hydration and swelling was investigated in [16], and a swelling model of the SC is proposed in [20]. Therefore, in the final skin model, the SC layer employs hydration dependency for the dielectric parameters and thickness. As previous work showed, the deeper skin layers cannot be neglected [10,14,22].

### 2.1. Stratum Corneum Hydration Model

This section introduces the stratum corneum model, as proposed by [20]. The final skin model consists of the SC model with no water concentration gradient, as described in Section 2.1.2. The dynamic SC model was used to investigate the hydration and dehydration process and to reproduce measurement steps from previous work for data utilized in this project.

#### 2.1.1. Dynamics of Water Transport in SC

The hydration model of the stratum corneum describes, on one hand, the hydration profile and the water gradient, and on the other hand, the change in the thickness (swelling and shrinking) with varying hydration. The convection–diffusion PDE with a moving boundary sufficiently describes the swelling model of the SC, that is,
(1)$$\frac{\partial C_{w}}{\partial t}=-v\frac{\partial C_{w}}{\partial z}+\frac{\partial }{\partial z}\left(D\frac{\partial C_{w}}{\partial z}\right),$$
with a water flux
(2)$$f=-D\frac{\partial C_{w}}{\partial z}+vC_{w},$$
where $C_{w}$ is the water concentration in $g/cm^{3}$, *v* the convective velocity in $g/(cm^{2}s)$, and *D* is the diffusivity at depth in $cm^{2}/s$.

The convective velocity is caused by the flux at the lower boundary. It can vary with time, but is not depth-dependent because it is assumed that the water partial molar volume in the SC is equal to the pure water molar volume. We have
(3)$$v=\frac{f_{0}}{\rho_{w}},$$
where $f_{0}=f\left(z=0\right)$ is the flux at the lower boundary and $\rho_{w}=1.0\;g/cm^{3}$ is the water density. The diffusivity
(4)$$D=\left(0.4331-0.3765\exp\left(-9.6215C_{w}\right)+0.00006428\exp\left(12.873C_{w}\right)\right)\times 10^{-9}$$
depends on $C_{w}$, as proposed in [19].

The moving boundary describes the swelling and shrinking of the SC, where its thickness δ changes with time. The rate of change is defined by the difference of the boundary fluxes, again with the assumption that the water partial molar volume in the SC is equal to the pure water molar volume.
(5)$$\frac{\partial \delta }{\partial t}=\frac{f_{0}-f_{\delta }}{\rho_{w}},$$
where $f_{0}=f\left(z=0\right)$ is the flux at the lower boundary and $f_{\delta }=f\left(z=\delta \right)$ is the flux at the skin surface.

If the skin is in contact with air, the flux at the upper boundary is dependent on the relative ambient humidity, and can be described by
(6)$$f_{\delta }=k_{g}\frac{\left(a_{w}-RH\right)p_{\mathrm{sat}}^{\circ }MW}{RT},$$
where $k_{g}=0.318\;cm\;s$^−1^ is the mass transfer coefficient [23], $a_{w}$ the water activity at the skin surface, RH the relative ambient humidity, $p_{\mathrm{sat}}^{\circ }=4.76\;\mathrm{kPa}$ is the water-saturated vapor pressure at skin surface temperature T=(32+273.2)K, MW=18 g mol^−1^ is water molecular weight, and $R=8.314\frac{m^{3}\mathrm{Pa}}{K\mathrm{mol}}$ is the ideal gas constant.

#### 2.1.2. SC Equilibrium Water Concentration

In the absence of a water activity gradient, the equilibrium water sorption volume *V* (g water/g dry tissue) can be determined by the Guggenheim–Anderson–deBoer (GAB) isotherm
(7)$$\frac{V}{V_{m}}=\frac{cka_{w}}{\left(1-ka_{w}\right)\left(1-ka_{w}+cka_{w}\right)},$$
with $a_{w}$ as the water activity, c=4.39, k=1/1.01, and $V_{m}=0.0386$ (g water/g dry tissue) [24]. The associated water concentration $C_{w}$ is related to *V* according to Equation (Equation 8)
(8)$$C_{w}=\frac{\rho_{w}\rho_{\mathrm{mem}}V}{\rho_{w}+\rho_{\mathrm{mem}}V},$$
with $\rho_{w}=1.0\;g/cm^{3}$ as the water density and $\rho_{\mathrm{mem}}=1.3\;g/cm^{3}$ as the dry tissue density [24]. With the assumption of constant partial molar volume for water, the equilibrium thickness of the SC δ can then be determined as
(9)$$\delta =\frac{\delta_{\mathrm{dry}}}{1-\frac{C_{w}}{\rho_{w}}},$$
where $\delta_{\mathrm{dry}}$ is the dry tissue thickness in μm, which was determined in [20] from the fully hydrated SC thickness of 43.4 μm [25].

### 2.2. Dielectric Properties of Skin

Biological tissue is a very heterogeneous material, and various components influence the dielectric properties, such as concentration of free water, bound water, and molecular compositions. The dielectric dispersion characteristics (or Debye-type dispersion) of biological tissues are commonly approximated by a Cole–Cole model [7,21,22,26]. The human skin is employed as a multilayer model due to separable dispersive dielectric behavior. In this model, the focus was on three layers; the outermost layer, the stratum corneum, the viable skin, which combines living epidermis and dermis, and the hypodermis as subcutaneous fat. Each skin layer is represented by a complex relative permittivity $\varepsilon_{r}=\varepsilon^{\prime }-j\varepsilon^{\prime \prime }$ characterized by the Cole–Cole model:(10)$$\varepsilon_{r}=\varepsilon_{\infty }+\sum_{n}\frac{\Delta \varepsilon_{n}}{1+{\left(j\omega \tau_{n}\right)}^{\left(1-\alpha_{n}\right)}}+\frac{\sigma_{\mathrm{DC}}}{j\omega \varepsilon_{0}},$$
where j is the imaginary unit, $\varepsilon_{\infty }$ is the optical permittivity, *n* is the number of dispersions, $\Delta \varepsilon_{n}$ is the *n*th permittivity increment, $\tau_{n}$ is the *n*th relaxation time, $\alpha_{n}$ is the *n*th broadening parameter, ω is the angular frequency, $\sigma_{\mathrm{DC}}$ is the static conductivity, and $\varepsilon_{0}$ is the dielectric permittivity of the vacuum.

The frequency-dependent relationship between the conductivity and relative permittivity is
(11)$$\sigma =j\omega \varepsilon_{0}\varepsilon_{r},$$
where the conductivity σ and the relative permittivity $\varepsilon_{r}$ are complex quantities [26].

#### 2.2.1. SC Dielectric Hydration Dependency

In [21], it was found that bound water has a negligible influence on the dielectric properties of the SC, and recommends simplifying the model to
(12)$$\varepsilon_{r}=\varepsilon_{\infty }+\frac{\Delta \varepsilon_{\mathrm{free}}}{1+j\omega \tau_{\mathrm{free}}}+\frac{\Delta \varepsilon_{\mathrm{slow}}}{1+j\omega \tau_{\mathrm{slow}}},$$
where the *free* parameters are caused by the free water in SC and the *slow* parameters are assumed to be caused by protein polarization.The optical permittivity directly depends on the permittivity increment of free water
(13)$$\varepsilon_{\infty }=3.3+\frac{2}{73.2}\Delta \varepsilon_{\mathrm{free}}.$$

The value 3.3 in (Equation 13) represents the relative permittivity of dry SC, which is a typical value for the dry protein [27], and the value 73.2 is the relative permittivity of free water [21].

As stated in [21], the dielectric properties of the SC vary with the hydration of the skin. Further, a linear dependency between the free water content of the SC and the permittivity increment of the free water $\Delta \varepsilon_{\mathrm{free}}$ was found. With the assumption that all dielectric parameters from (Equation 12) undergo the same linear dependency, the hydration relation can be fitted with linear regression for each parameter. To relate the dielectric parameter directly to the water activity, the water concentration $C_{w}$ is formulated in dependency of the water activity $a_{w}$ using the Equations (Equation 7) and (Equation 8).

We obtain
(14)$$C_{w}\left(a_{w}\right)=\frac{V_{m}ck\rho_{\mathrm{mem}}a_{w}}{1+\left(V_{m}ck\frac{\rho_{\mathrm{mem}}}{\rho_{w}}-2k+ck\right)a_{w}+\left(k^{2}-ck^{2}\right)a_{w}^{2}},$$
where $C_{w}$ is the resulting water concentration in the SC by the given water activity $a_{w}$. Applying the linear regression to the dielectric parameters, the equation becomes:(15)$$\vec{y}=\beta_{0}+\beta_{1}\vec{C_{w}},$$
with $\vec{y}$ as the sample vector representing the dielectric parameters, $\vec{C_{w}}$ is the vector of water concentrations at measurement of the dielectric parameters, and $\beta_{0}$ and $\beta_{1}$ are the parameters to be estimated. The linear regression fit resulted in a scaling of $C_{w}$ (with $\beta_{0}$ set to 0) because the provided measurements only represent the hydrated state of the SC, and no samples for dry tissue are given. The samples and fit parameters are listed in Table 1. For a complete derivation of the fit, see Appendix A.

The interpolation equations take the form
(16)$$\begin{array}{cc} \Delta {\hat{\varepsilon }}_{\mathrm{free}} & =\beta_{\Delta \varepsilon_{\mathrm{free}}}C_{w}\left(a_{w}\right), \end{array}$$
(17)$$\begin{array}{cc} {\hat{\tau }}_{\mathrm{free}} & =\beta_{\tau_{\mathrm{free}}}C_{w}\left(a_{w}\right), \end{array}$$
(18)$$\begin{array}{cc} \Delta {\hat{\varepsilon }}_{\mathrm{slow}} & =\beta_{\Delta \varepsilon_{\mathrm{slow}}}C_{w}\left(a_{w}\right), \end{array}$$
(19)$$\begin{array}{cc} {\hat{\tau }}_{\mathrm{slow}} & =\beta_{\tau_{\mathrm{slow}}}C_{w}\left(a_{w}\right), \end{array}$$
where each dielectric parameter has its own scaling parameter β. The quantities $\Delta {\hat{\varepsilon }}_{\mathrm{free}}$, ${\hat{\tau }}_{\mathrm{free}}$, $\Delta {\hat{\varepsilon }}_{\mathrm{slow}}$, and ${\hat{\tau }}_{\mathrm{slow}}$ stand for the interpolated dielectric parameters.

The final estimated relative permittivity for the SC is formulated as
(20)$${\hat{\varepsilon }}_{r}={\hat{\varepsilon }}_{\infty }+\frac{\Delta {\hat{\varepsilon }}_{\mathrm{free}}}{1+j\omega {\hat{\tau }}_{\mathrm{free}}}+\frac{\Delta {\hat{\varepsilon }}_{\mathrm{slow}}}{1+j\omega {\hat{\tau }}_{\mathrm{slow}}}.$$

### 2.3. Electrode Model

For dielectric spectroscopy, sensors with different electrode geometries and different electrode setups are available (e.g., multiple electrode pairs, one sensing multiple driving, interdigitated fingers). These sensors can be categorized according to their measurement principle as a conductance sensor and capacitance sensor. For the conductance measurement, the electrodes are in direct contact with the skin, whereas for the capacitance measurements, the sensors have an insulating layer between the electrodes and the skin [28].

The dimensions of the electrodes and the skin are much smaller than the smallest wavelength of the applied signal. Therefore, a quasi-static approximation of Maxwell’s equations can be applied with $\nabla \times \vec{E}=0$. The capacitance *C* of the skin can be calculated by solving the current conservation equations
(21)$$\begin{array}{cc} \nabla \cdot \vec{J} & =Q, \end{array}$$
(22)$$\begin{array}{cc} \vec{J} & =\sigma \vec{E}+j\omega \vec{D}+\vec{J_{e}}, \end{array}$$
(23)$$\begin{array}{cc} \vec{E} & =-\nabla V. \end{array}$$

Here, $\vec{J}$ is the current density, *Q* the charge density, $\vec{E}$ the electric field, $\vec{D}=\varepsilon_{r}\varepsilon_{0}\vec{E}$ the electric displacement field with $\varepsilon_{r}$ and $\varepsilon_{0}$ as the relative permittivity and the permittivity in vacuum, $\vec{J_{e}}$ the external current density, *V* the applied electric potential, σ the electric conductivity of the material, and ω=2πf is the angular velocity.

### 2.4. Numerical Analysis

For the numerical computation of the above-listed models, the finite-element method (FEM)-based simulation software COMSOL Multiphysics^®^ version 5.6 (COMSOL Multiphysics GmbH, Zurich, Switzerland) was used.

As explained in Section 2.2.1, the dielectric parameters for the SC layer are interpolated with Equation (Equation 16). In [22], the epidermis/dermis layer was approximated by blood and the hypodermis by infiltrated fat. The dielectric parameters for these two layers were taken from [7], as listed in Table 2.

For the evaluation of the water concentration in the SC and the reproduction of the measurements in [21], the dynamics of water transport in the SC model was used (Equation 1)–(Equation 6). The SC was initialized as fully hydrated and exposed to a relative humidity of 30% until equilibrium was reached. The 30% of relative humidity was chosen to simulate a realistic dry environment. The reached equilibrium then served as initial SC hydration for the simulation of the hydration boundaries and other hydration levels, as investigated in Section 3.1. Table 3 summarizes the initial conditions and boundary conditions.

## 3. Results and Discussion

Under the assumption that the model resembles realistic dispersive dielectric skin behavior, various influences on the electrodes’ measurements were investigated. The skin model was dimensioned (width, height) so that the boundaries of electric insulation $\vec{n}\vec{J}=0$ did not influence more than 1%, where $\vec{n}$ is the normal vector of the boundary.

As mentioned in Section 2.3, different types of fringing-field electrodes are available. In this project, three different electrodes were used: the circular conductance electrode, interdigitated conductance electrode, and interdigitated capacitance electrode. The layered model and the electrode geometries are illustrated in Figure 1.

### 3.1. Range of Hydration Measurement

For a better insight of the stratum corneum water concentration range under conditions without special influences (i.e., no sweating, no skin disease, no wounds), the boundaries were evaluated by exposing the SC surface to air with a humidity level of 0% and 100% by means of the dynamics of water transport in SC (Equation 1)–(Equation 6). The simulation resulted in a water concentration of $C_{w}=0.585\;g/cm^{3}$ at dry air with 0% relative humidity and $C_{w}=0.781\;g/cm^{3}$ at humid air with 100% relative humidity, as shown in Figure 2. A linear relation between dielectric relaxation strength and free water content for water concentration above 0.45 g/cm^3^ is proposed in [21]. The measurement range according to the simulation is clearly above 0.45 g/cm^3^ and within the linear relation. This justifies the linear approximation of the dielectric parameters in Section 2.2. Additionally for this reason, the following investigations were performed in the high water concentration range.

### 3.2. Influence of Varying Stratum Corneum Thickness on Measurement

With the changing dielectric SC properties depending on the water content, the question arises as to what influence the SC thickness, also depending on the water content, has on the measurement. The influence of the varying thickness has been investigated per electrode type at a measurement frequency of 1 MHz and compared to the results with constant thickness. In Figure 3, the simulated capacitance of the three different electrode types is displayed in relation to the water activity $a_{w}$.

Figure 3 indicates that for all electrode types with varying thicknesses, the capacitance first increases with increasing water activity and drops again at water activity $a_{w}>0.95$. Further, with constant thickness, the capacitance increases with increasing water activity.

The following points must be considered when interpreting the simulated effects of the comparison of skin with SC swelling with skin with constant thickness. The major component of simulated capacitance is due to the electric field that penetrates into the epidermis/dermis layer because it has higher relative permittivity than SC, see Table 1 and Table 2. If the SC is swelling due to hydration, the electric field penetrates less deep into the epidermis/dermis, and hence there is less capacitance.

The increase of the capacitance for water activity $a_{w}<0.95$ and a decrease for $a_{w}>0.95$ can be hypothesized due to the non-linear swelling of the SC, as seen in Figure 4. For water activity $a_{w}<0.95$, the SC thickness varies less than 5 μm. The SC relative permittivity increases with the water activity, as explained in Section 2.2.1. Therefore, for water activity $a_{w}<0.95$, the increase of the capacitance due to increasing relative permittivity is bigger than the reduction of the capacitance due to increasing SC thickness. Due to the non-linearity of the SC swelling, the SC thickness increases about 30 μm in water activity $a_{w}>0.95$. In that high water activity range, the reduction of the capacitance due to SC thickness is stronger than the increase from the SC relative permittivity. Therefore, a decrease in capacitance is observed compared to lower water activities.

The magnitude difference between the variable and constant thickness is again due to the influence of the epidermis/dermis layer. At constant thickness (here at dry thickness 9.55 μm), the electric field propagates deeper into the epidermis/dermis than with variable thickness, and therefore results in higher capacitance.

According to this model, varying the thickness depending on the hydration level has a significant impact on the calculated capacitance that cannot be neglected. The penetration depth of the electric field could also be investigated by increasing the SC thickness until the capacitance converges and the influence of the epidermis becomes negligible.

#### Influence of Varying Stratum Corneum at Various Frequencies

Figure 5 shows the influence of the varying thickness at logarithmically spaced frequencies from 10 kHz to 1 GHz. It is visible that with increasing frequency, the magnitude of the simulated capacitance decreases. The trend of the capacitance with varying water activity due to SC swelling has an effect throughout all frequencies with a turning point at $a_{w}∼0.95$.

This effect is as expected. The drop in the magnitude of the capacitance with increasing frequency can be explained with the SC dielectric spectrum, see Appendix A. In the frequencies of 1 MHz and lower, the SC relative permittivity is dominated by bound water and protein polarization. At frequencies above 1 MHz, the β-dispersion occurs and the relative permittivity is increasingly more dominated by free water.

### 3.3. Electrode Dimension Sensitivity

The sensitivity analysis was performed at different SC hydration levels within a water activity range of $a_{w}=0.8$ to 0.996 at 1 MHz on the three fringing-field electrode types, that is, the circular conductance electrode, interdigitated conductance electrode, and interdigitated capacitance electrode.

For the analysis, we define the relative capacitive electrode sensitivity as
(24)$$r=\frac{C_{a_{w}}-C_{0.996}}{C_{0.996}},$$
where $C_{a_{w}}$ is the simulated capacitance in F at specified water activity $a_{w}$ and $C_{0.996}$ is the simulated capacitance in F at water activity $a_{w}=0.996$. For each electrode, the relative capacitive sensitivity was calculated with Equation (Equation 24) in relation to the simulated capacitance at the fully hydrated level of $a_{w}=0.996$. The influence of the electrode dimensions was investigated by varying the size of the electrodes and the gap between the electrodes. The areas of the driving and sensing electrode were always kept equal.

In Figure 6, it is visible that when reducing both the electrode gap and the electrode size of the circular conductance electrode, the sensitivity increases. In Figure 7a, the sensitivity of the interdigitated conductance electrode reaches an optimum at an electrode gap close to 0.12 mm. In Figure 7b, however, it is visible that the sensitivity increases with increasing electrode width. As can be seen in Figure 8, the interdigitated capacitance electrode shows two different sensitivity ranges with the border at $a_{w}∼0.95$. For both electrode types, the sesitivity increases with increasing electrode gap and width at high water activity of $a_{w}>0.95$. At low water activity $a_{w}<0.95$, however, the electrodes with wider gaps are more sensitive. In contrary, in the same low water activity range, the smaller the electrode width, the more sensitive.

Comparing the values of three electrodes’ relative sensitivity, in the hydration range $a_{w}<0.95$, the sensitivity values of the conductance electrodes are 10 times higher than the sensitivity values of the capacitance electrode. In the higher hydration range with $a_{w}>0.95$, the sensitivity of the conductance electrodes is even up to a multiple of 100 times higher. Thus, the sensitivity of the conductance electrodes is more strongly affected by a geometry change than the capacitance electrode.

The current density of the circular conductance electrode in Figure 9a indicates that the charge exchange mainly occurs at the edge between the electrodes. Therefore, it is assumed that by reducing the electrode dimensions, the charge density increases, and with it, the sensitivity. However, this hypothesis is contradictory to what is observed in the sensitivity analysis of the interdigitated conductance electrode, where the sensitivity increases by increasing the dimensions.

An optimum of the electrode becomes visible in Figure 7a. However, for the other geometry sweeps, neither an optimum nor a convergence in the sensitivity is found. The optimum might be out of the examined geometry range or only exists at zero or infinity.

This work will enable better electrode designs and will be beneficial to several applications. In the fields of cosmetics and drug delivery, it will allow to improve upon current methods used to investigate and characterize SC hydration [29,30]. For example, it has been experimentally reported that some types of electrodes exhibit better performances when measuring either very hydrated or very dry skin, without providing theoretical explanations for such results [31,32]. In the field of lymphatic diseases, diagnostic, electrode-based sensors are used to measure the excess amount of water in the dermis that correlates with an impaired lymphatic function [2,5]. Thanks to this study, new electrodes specifically designed to monitor water content in the deeper skin layers could be developed. Sensitivity analysis with respect to electrode dimensions can be applied to optimize the stability of stretchable electrodes for wearable hydration sensors [4,5,6]. Another application area is diagnosing malignant skin tumors [3]. Finally, such a computer model will be useful for the design of reliable EEG electrodes, especially semi-dry, where skin hydration impacts the measurements [33,34,35].

## 4. Conclusions

In this work, we reported a computational model simulating the human skin for dielectric spectroscopy applications. The skin is modelled as a multilayer system, each layer exhibiting different dielectric dispersive parameters. In addition, the outer skin layer, i.e., the stratum corneum, implements a swelling model with hydration-dependent thickness.

The exposition of the skin surface to dry air with ambient relative humidity of 0% and humid air with ambient relative humidity of 100% allowed to retrieve the physiological ranges of water concentration and water activity. With a water concentration between 0.585 g/cm^3^ and 0.781 g/cm^3^, or expressed in term of water activity between 0.9745 and 0.996, the stratum corneum is always fully hydrated.

The more the electric field penetrates into the epidermis/dermis layers, the higher the resulting capacitance due to the high dispersive dielectric parameters of those layers in comparison to the stratum corneum. Varying the stratum corneum thickness with hydration impacts the penetration depth into the lower skin layers. At high water activity, a thicker stratum corneum reduces the penetration depth of the electric field into the epidermis/dermis layer and as a result, the capacitance decreases.

In addition, we examined the sensitivity of three different fringing-field electrode types. The analysis demonstrated that, as expected, the sensitivity is geometry and dimension dependent. Nonetheless, as conductance electrodes exhibit two orders of magnitude higher sensitivity, they are more affected by a change in dimension than capacitance electrodes.

The presented model opens many potential applications in terms of electrode design optimization or to provide a deeper understanding of the variations observed experimentally when using different skin hydration sensors.

## Figures and Tables

**Figure 1 sensors-21-03986-f001:**
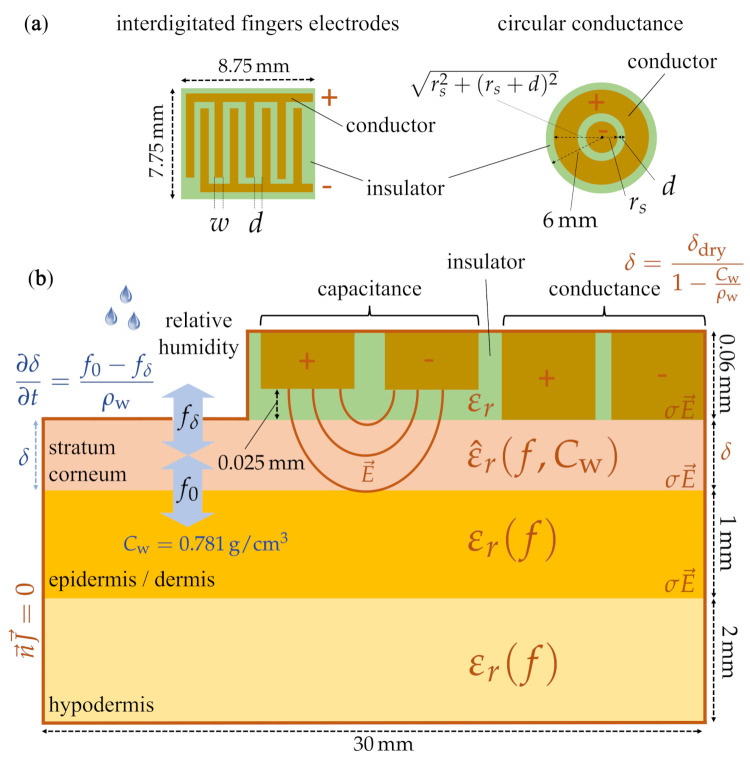
Pictorial description of the numerical model, with dimensions and boundary conditions, developed in this work to investigate the electrode-based stratum corneum hydration measurement. (**a**) Interdigitated fingers electrode (**left**) and circular conductance electrode (**right**) with the outer driving and inner sensing electrodes and their respective dimensions. In this study, *w*, *d* and $r_{s}$ were varied to investigate the effect of the electrode dimensions on the electrode sensitivity. (**b**) Skin model. The skin is modelled as a multilayer material (stratum corneum, epidermis/dermis and hypodermis). The thickness, as well as the dielectric properties of the stratum corneum varies with hydration, whereas properties of other layers are taken constant. We assume discontinuity of the electric field between the different layers due to different dielectric properties. The skin is insulated at its outer borders ($\vec{n}\vec{J}=0$). The conductance electrode (in electrical contact with the skin) and capacitance electrode (insulation layer between the electrode and the skin) are both investigated in this work. The electric field results from a potential difference between the two electrodes.

**Figure 2 sensors-21-03986-f002:**
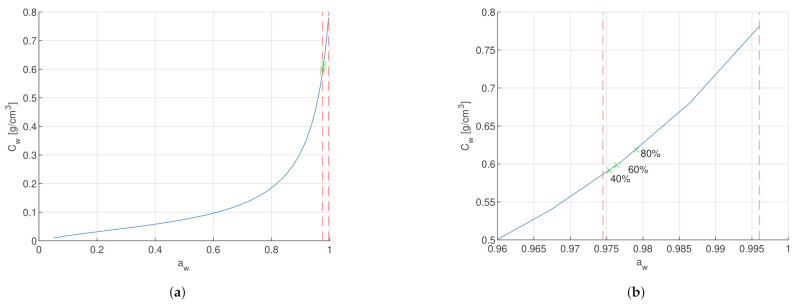
Water concentration and water activity relationship. (**a**) Blue line is the water concentration *C_w_* in relation to the water activity *a_w_*. The red dashed line on the left marks the water activity at ambient relative humidity of 0% at *a_w_* = 0.9745 according to Equations (Equation 7) and (Equation 8). The red dashed line on the right marks the water activity at ambient relative humidity of 100% at *a_w_* = 0.996, which is equal to fully hydrated SC. (**b**) Zoom of the water concentration *C_w_* in relation to the water activity *a_w_* into the marked range. The three green marks on the blue line indicate the resulting water concentration of the SC exposed to ambient relative humidity of 40%, 60%, and 80% for 30 min, see Appendix A.

**Figure 3 sensors-21-03986-f003:**
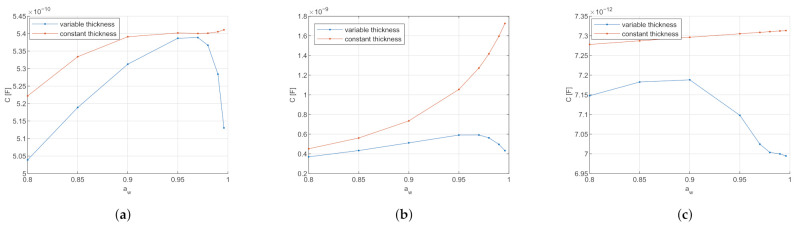
Comparison of electrode capacitance for constant thickness (in red) and humidity-dependent stratum corneum thickness (in blue). The dimensions of the skin are for epidermis/dermis 1 mm and for hypodermis 2 mm. The dry SC thickness in (Equation 9), as well as the constant thickness is set to 9.55 μm. (**a**) Comparison for the circular conductance electrode: The influence of the varying thickness is especially visible in the higher water activity range, where the capacitance reduces with varying thickness but increases with constant thickness. (**b**) Comparison for the interdigitated conductance electrode: The influence of the varying thickness is visible over the complete water activity range. With constant thickness, the capacitance grows exponentially over the full range, whereas with varying thickness, the capacitance increases linearly at lower water activity and decreases at high water activity. (**c**) Comparison for the interdigitated capacitive electrode: the influence of the varying thickness is visible over the complete water activity range. With constant thickness, the capacitance grows linearly over the full range, whereas with varying thickness, the capacitance first increases at lower water activity and then decreases at high water activity.

**Figure 4 sensors-21-03986-f004:**
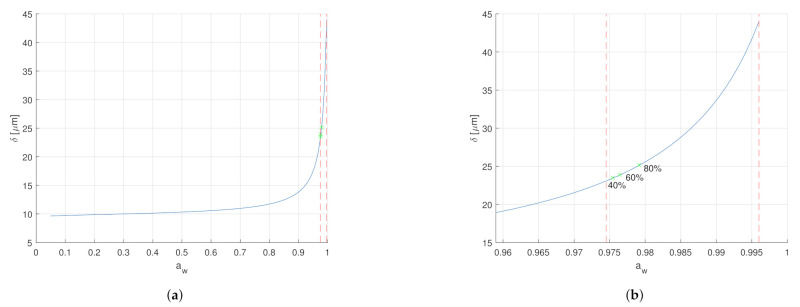
SC thickness and water activity relationship. (**a**) The blue line indicates the SC thickness in relation to the water activity according to Equations (Equation 7)–(Equation 9). The red dashed line on the left marks the water activity at ambient relative humidity of 0% at *a_w_* = 0.9745. The red dashed line on the right marks the water activity at ambient relative humidity of 100% at *a_w_* = 0.996, which is equal to fully hydrated SC. The dry SC thickness is 9.55 μm. The non-linearity is clearly visible. In the low water activity range *a_w_* < 0.95, the SC thickness only grows approximately 5 μm. However, in the higher water activity range *a_w_* < 0.95, the SC thickness grows by approximately 30 μm. (**b**) Zoom of the SC thickness d in relation to the water activity *a_w_* into the marked range. The three green marks on the blue line indicate the resulted thickness of the SC exposed to ambient relative humidity of 40%, 60%, and 80% for 30 min, see Appendix A.

**Figure 5 sensors-21-03986-f005:**
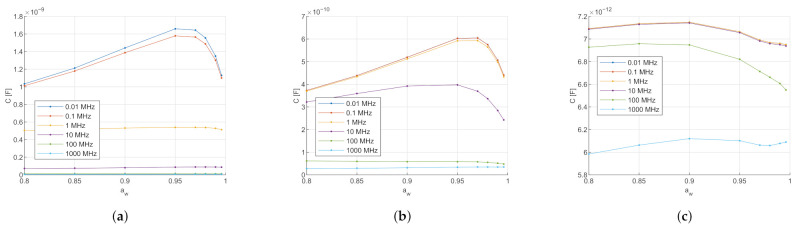
Simulated capacitance of the three electrode types at logarithmically spaced frequencies from 10 kHz to 1 GHz. The dimensions of the skin are for the epidermis/dermis 1 mm and the hypodermis 2 mm. The dry SC thickness in (Equation 9) is set to 9.55 μm. The dispersive dielectric behavior is visible with the capacitance reducing with increasing frequency. (**a**) Circular conductance electrode. (**b**) Interdigitated conductance electrode. (**c**) Interdigitated capacitance electrode.

**Figure 6 sensors-21-03986-f006:**
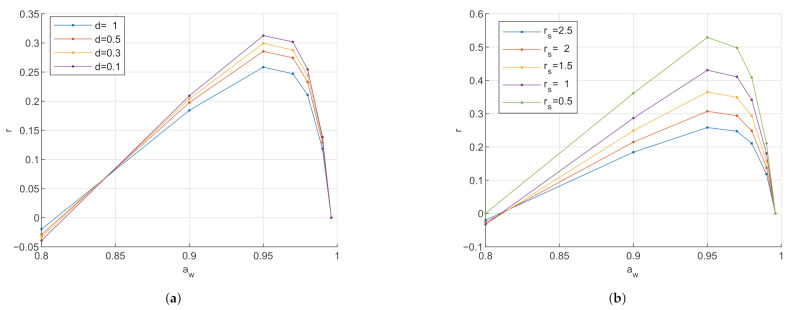
Relative sensitivity analysis of the circular conductance electrode at 1 MHz. (**a**) Relative sensitivity *r* with varying electrode gap: Parameter *d* is the electrode gap in mm. The radius of the sensing electrode was kept at 2.5 mm. The radius of the driving electrode was adjusted to keep the area equal to the sensing electrode area. Reducing the electrode gap increases the sensitivity. (**b**) Relative sensitivity *r* with varying electrode radius: Parameter *r_s_* is the radius of the sensing electrode in mm. The electrode gap was kept at 1 mm. The radius of the driving electrode was adjusted to keep the area equal to the sensing electrode area. Reducing the electrode area increases the sensitivity.

**Figure 7 sensors-21-03986-f007:**
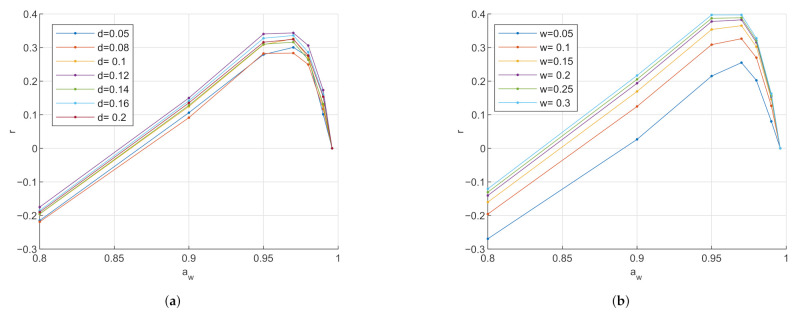
Relative sensitivity analysis of the interdigitated conductance electrode at 1 MHz. (**a**) Relative sensitivity *r* with varying electrode gap: Parameter *d* is the electrode gap between the electrode fingers in mm. The electrode width was kept at 0.1 mm. An optimum in sensitivity is reached with an electrode gap at *d* ∼ 0.12 mm. (**b**) Relative sensitivity *r* with varying electrode size: Parameter *w* is the electrode width of the electrode fingers in mm. The electrode gap was kept at 0.1 mm. Increasing the electrode width increases the sensitivity.

**Figure 8 sensors-21-03986-f008:**
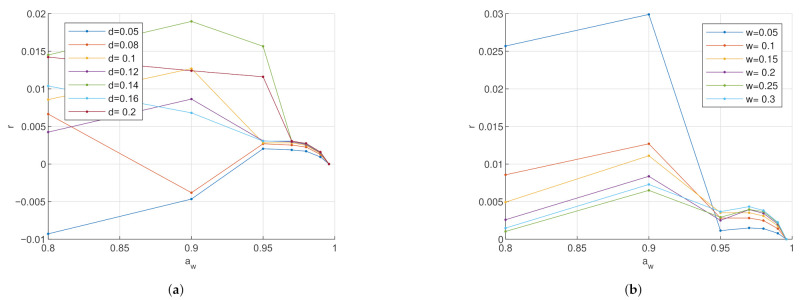
Relative sensitivity analysis of the interdigitated capacitive electrode at 1 MHz. (**a**) Relative sensitivity r with varying electrode gap: Parameter *d* is the electrode gap between the electrode fingers in mm. The electrode width was kept at 0.1 mm. Increasing the electrode gap increases the sensitivity. (**b**) Relative sensitivity *r* with varying electrode size: Parameter *w* is the electrode width of the electrode fingers in mm. The electrode gap was kept at 0.1 mm. In a higher water activity range *a_w_* > 0.95, the wider the electrode, the better the sensitivity. In a lower water activity range *a_w_* < 0.95, the smaller the electrode width, the better the sensitivity.

**Figure 9 sensors-21-03986-f009:**
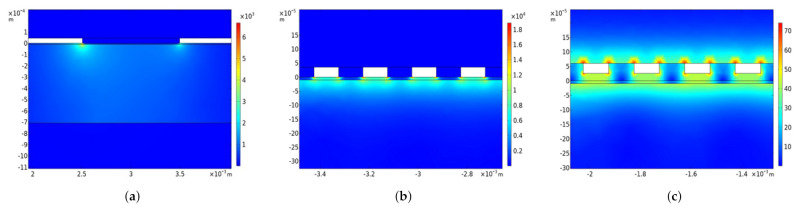
Illustration of the current density propagation. The X and Y axes are the size measures of the simulation in m. The color gradient shows the current density in Am^−2^. (**a**) Current density propagation through the skin and inside the circular conductance electrode: The charges penetrate vertically through the SC and move mainly in the epidermis/dermis due to higher conductivity. It is also visible that the charge exchange occurs in the edge between the electrodes and almost no charge leaves the outer surface. (**b**) Current density propagation through the skin and inside the interdigitated conductance electrode: The charges penetrate vertically through the SC and move mainly in the epidermis/dermis due to higher conductivity. (**c**) Current density propagation through the skin and inside the interdigitated capacitance electrode: A significant amount of the charges move inside the electrode due to the insulator layer between electrode and skin. The charges that penetrate into the skin again move vertically through the SC and connect in the epidermis/dermis due to higher conductivity.

**Table 1 sensors-21-03986-t001:** Cole–Cole parameters from [21] at three hydration levels were used to approximate the SC hydration dependency. β lists the calculated scaling value resulting from the linear regression. $\varepsilon_{\infty }=3.3+\frac{2}{73.2}\Delta \varepsilon_{\mathrm{free}}$ can be calculated from $\Delta \varepsilon_{\mathrm{free}}$.

Parameter	RH=40%	RH=60%	RH=80%	β
$\Delta \varepsilon_{\mathrm{free}}$	31.5	33.2	35.9	0.0557
$\tau_{1}$ [ps]	8	10	10	$1.551\times 10^{-14}$
$\Delta \varepsilon_{\mathrm{slow}}$	156	250	370	0.432
$\tau_{2}$ [ns]	6	9	17	$1.785\times 10^{-11}$

**Table 2 sensors-21-03986-t002:** Cole–Cole parameters for epidermis/dermis (E/D) and the hypodermis (HYP) layer in the skin model. The values refer to blood for E/D and infiltrated fat for HYP from [7].

Parameter	E/D	HYP
$\sigma_{\mathrm{DC}}$ [S/m]	0.7	0.035
$\varepsilon_{\infty }$	4	2.5
$\Delta \varepsilon_{1}$	56	9
$\tau_{1}$ [ps]	8.38	79.9
$\alpha_{1}$	0.1	0.2
$\Delta \varepsilon_{2}$	5200	35
$\tau_{2}$ [ps]	132.6	15.92
$\alpha_{2}$	0.1	0.1
$\Delta \varepsilon_{3}$	-	33,000
$\tau_{3}$ [ps]	-	159
$\alpha_{3}$	-	0.05
$\Delta \varepsilon_{4}$	-	$10^{7}$
$\tau_{4}$ [ps]	-	15.9
$\alpha_{4}$	-	0.01

**Table 3 sensors-21-03986-t003:** Initial conditions (IC) and boundary conditions (BC) used to evaluate the water concentration in the SC. The inner boundary is connected to the epidermis and is assumed to always be fully hydrated $C_{w}=0.781\;g/cm^{3}$. The initial conditions for the hydration steps are the resulting equilibrium of the first step to simulate normally hydrated skin. During all steps, the skin is exposed to air. Therefore, the flux $f_{\delta }$ is dependent on the ambient relative humidity according to (Equation 6).

Step	IC	BC at z=0	BC at z=δ
(1) Dehydration	$C_{w}=0.781\;g/cm^{3}$	$C_{w}=0.781\;g/cm^{3}$	RH=0.3, $f_{\delta }$
(2) Dehydration	result of step (1)	$C_{w}=0.781\;g/cm^{3}$	RH=0, $f_{\delta }$
(3) Hydration	result of step (1)	$C_{w}=0.781\;g/cm^{3}$	RH=0.4, $f_{\delta }$
(4) Hydration	result of step (1)	$C_{w}=0.781\;g/cm^{3}$	RH=0.6, $f_{\delta }$
(5) Hydration	result of step (1)	$C_{w}=0.781\;g/cm^{3}$	RH=0.8, $f_{\delta }$
(6) Hydration	result of step (1)	$C_{w}=0.781\;g/cm^{3}$	RH=1, $f_{\delta }$

## Data Availability

Not applicable.

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
