# Peer review of "Modeling Stratum Corneum Swelling for the Optimization of Electrode-Based Skin Hydration Sensors"

_sensors, 2021, doi:10.3390/s21123986_

Round 1
Reviewer 1 Report
The authors model stratum corneum swelling for the optimization of electrode-based skin hydration sensors. The work have great significance in the related field. Although being interesting, I find that there are some major issues with the paper that require addressing prior to this being considered for publication in this journal.
This manuscript is full of spelling typos, style errors and grammatical errors, which severely affect its readability. Pleases carefully check and correct them in the revised manuscript. I recommend the authors add a representative scheme for deeply undstanding their idea. Indeed, the skin hydration is very important for design reliable EEG electrodes, espcically for semi-dry electrodes. So I hope the authors disscuss this point in the disscussion section. In addition, some ralated references are recommended to be cited in the revised manuscript, including but not limiting to J. Neural Eng. 17 (2020) 026001; J. Neural Eng. 17 (2020) 051004; J. Neural Eng. 18 (2021) 046016.
Author Response
The authors would like to thank the reviewer for the comments that helped us improving the manuscript. All the changes have been highlighted in the revised version.
We improved the manuscript readability by correcting typos, style errors and grammatical errors. The revised manuscript has been reviewed by a native English speaker.
Following reviewer recommendation, we add a new figure (Figure 1) summarizing the concept proposed in this work.
Finally, a new paragraph in the “Results and Discussion” section has been added discussing potential applications of the model presented in this study and cites new references including the ones proposed by the reviewer.
Reviewer 2 Report
The authors are invited to address the following points:
1) What is the significance of section 2.1.1? If these equations are never used in simulations, what is the point of presenting them in the main text? It looks like only eqs (7) and (8) are used from transport point of view. Only eq. (6) is mentioned in table (3). Please explain.
2) Please give the frequency range of the study in section 2.2. At our given freq range (I believe it is 10 kHz to 1 GHz) what are the dielectric dispersion mechanisms? Please write them out and support with references. It looks like in the methodology the interfacial polarization (Maxwell Wagner effects) was never considered. Furthermore, in section 2.2.1 please explicitly mention that the dielectric properties are taken from another study. Please explain the fit procedure with few sentences and refer to the supplementary material for details. Is the DC conductivity included in the fit? I can't see it from the main text.
3) A distinction was made between impedance and capacitance electrodes. It might be scientifically more sound to use conductance and capacitance electrodes instead.
4) Please show the computational model in a schematic. Label and dimension all the parts of the skin, and show the boundary conditions at each interface.
Author Response
We would like to thank the reviewer for his/her constructive remarks about our model. All the changes have been highlighted in the revised version.
1) The model implementing the equations described in “2.1.1 Dynamics of Water Transport in SC” is used to analyze the hydration and dehydration process over time. It serves to identify the water concentration boundaries of the SC exposed to air at relative humidity of 0% and 100% as described in “3.1. Range of Hydration Measurement” and to reproduce the existing water concentration during the measurement of the dielectric properties of human SC at 40%, 60%, and 80% relative humidity.
The equations are now referenced in the main text for clarity.
2) The dielectric properties of the three layers were taken from literature, which are based on experimental measurements. Therefore, we assume that all occurring dispersion mechanisms are considered. The frequency ranges from 10Hz to 100GHz. Section “2.4 Numerical Analysis” explicitly mentions the literature source for the dielectric parameters and they are listed in table 2 as well.
Interfacial polarization within each layer (eg. caused at the cell membranes) is regarded by its dielectric spectrum. Further, due to different dielectric properties of the layers, the electric field is not continuous across the layer boundaries. However, as rightly pointed out by the reviewer, the effect of electrode polarization was neglected in this project.
The DC conductivity is not included in the fit because the model for the relative permittivity of SC was simplified according to the referenced literature, which also is the source of the data for dielectric properties of SC. The fit is explained in the main text, with reference to supplementary information.
3) We thank the reviewer for this remark, and we used conductance and capacitance distinctly in the revised manuscript.
4) As proposed, we added a new figure (Figure 1) describing the computational model with dimension of all parts and boundary conditions at each interface.
Reviewer 3 Report
The manuscript is overall very well written and the topic researched very thoroughly. The immediate results of the comparison of different electrodes used for skin impedance and capacitance measurements will be of interest for the developers of such technology. The insight gained in this study on the measureable capacitance and its dependence on SC water content and swelling can be expected to be of considerable interest to a broad audience of researchers in dermatology, pharmacology and skin research.
I do personally value the effort put into the collection of the data (experimental and computed) from the literature most of all, as I have not seen it compiled to this extent before.
The calculations and computational modeling are clearly presented as far as I can judge, being a biochemist myself.
I have a few suggestions for minor improvements, and spotted one spelling mistake:
Figure 3: in the Figure the graph can be seem to rise up to ~44 µm, that seems excessive from an experimenter’s point of view. Please comment and if possible, compare to experimental data from the literature (in the main text).
Most importantly: A zoomed in version of this figure as in Figure 2b is crucial to judge the difference in thickness which is relevant under ambient conditions (40-80% rH). In order to understandthe data, I had to build myself this image (see attached).
Please at “by” here, and delete the “final”: “In the low water activity range aw < 0.95, the SC thickness only grows by approximately 5μm.
However, in the higher water activity range aw < 0.95, the SC thickness grows by approximately 30μm”
Figure 4: (description) change thinkness to thickness
Figure 9: In the legend (next to the color gradient) the numbers are too small to read.

Author Response
The authors would like to thank the reviewer for his valuable suggestions and comments that helped us improving the quality of the manuscript. All the changes have been highlighted in the revised version.
1) The 44μm increase originates from reference (Wang et al. 2006). The authors are aware that slightly different values have been proposed in literature (for example in Crowther J.M., Matts P.J., Kaczvinsky J.R. (2012) Changes in Stratum Corneum Thickness, Water Gradients and Hydration by Moisturizers. In: Lodén M., Maibach H. (eds) Treatment of Dry Skin Syndrome. Springer, Berlin, Heidelberg. https://doi.org/10.1007/978-3-642-27606-4_38). This will be further investigated into in a follow-up study where we will analyze the implications of such findings on our model.
2) We agree with the reviewer that a zoom to judge the difference in thickness which is relevant under ambient conditions (40-80% rH) is crucial. We thank the reviewer for his/her inputs. We added such a zoom in Figure 3 (see revised manuscript).
3) We implemented the improvements proposed by the reviewer.
4) We apologize for this typo and corrected it as suggested by the reviewer.
5) We thank the reviewer for this comment and increased the font size in the legend of Figure 9.
Reviewer 4 Report
A VERY INTERESTNG BUT STRICTLY THEORETICAL APPROACH AS STATED ,,,,Different physical models are combined to simulate the behavior of the human skin under dielectric spectroscopy measurements,, DESCRIBING THE PROPERTIES/HYDRATION OF HUMAN SKIN
IN CONCLUSION THE AUTHORS STATE THAT ,,,,,The presented model opens many potential applications in terms of electrode design optimization or to provide a deeper understanding of the variations observed
experimentally when using different skin hydration sensors.
IT IS HIGHLY DESIRABLE THAT A NUMBER OF APPLICATIONS SHOULD BE MENTIONED AND WELL DOCUMENTED, I.E. THERE IS A GREAT INTEREST THESE DAYS TO EPLORE THE POSSIBLE SKIN DAMAGE MADE BY 5G HIGH FREQUENCY EXPOSURE. ANOTHER ISSUE TO CONSIDER IS THE RELATIONSHIP OF SAR WITH THE HYDRATION PROPERTIES UPON RADIATION EXPOSURE. THESE ISSUES COULD FILL THE GAP OF LACK OF A CLEAR SCOPE OF THE STUDY AND WOULD MAKE THE WORK PUBLISHABLE.
Author Response
The authors would like to thank the reviewer for his/her valuable inputs that helped us improving the quality of the manuscript. All the changes have been highlighted in the revised version.
We agree with the reviewer that the manuscript was lacking well documented applications. Therefore, we introduced a new paragraph with new references in the “Results and Discussion” section specifically discussing the potential applications of our model. If the issues raised by the reviewer (skin damage resulting to 5G exposure and hydration dependency upon radiation exposure) are definitely of interest, they do not really fit the scope of this paper dedicated to the investigation of skin hydration with electrodes in direct contact with the skin. Therefore, we focused on other potential applications in the cosmetics and diagnostics fields but we will consider reviewer’s comment for further investigation.
Round 2
Reviewer 1 Report
Accept in present form.